# Ginsenoside Rh1 Exerts Neuroprotective Effects by Activating the PI3K/Akt Pathway in Amyloid-β Induced SH-SY5Y Cells

**Miey Park** [1,2], **So-Hyeun Kim** [1] and **Hae-Jeung Lee** [1,2,*]

1   Department of Food and Nutrition, College of BioNano Technology, Gachon University, Seongnam-si 13120, Gyeonggi-do, Korea; mieyp@naver.com (M.P.); sohun94@naver.com (S.-H.K.)
2   Institute for Aging and Clinical Nutrition Research, Gachon University, Seongnam-si 13120, Gyeonggi-do, Korea
*   Correspondence: skysea@gachon.ac.kr; Tel.: +82-31-750-5968; Fax: +82-31-724-4411

**Abstract:** Alzheimer's disease (AD) is a neurodegenerative disorder characterized by the accumulation of β-amyloid plaques and hyperphosphorylated tau proteins in the brain. Cell signaling pathways such as PI3K/Akt are known to play an essential role in regulating cell survival, motility, transcription, metabolism, and progression of the cell cycle. Recent studies demonstrated that the disruption of these signaling pathways in neurodegenerative disorders leads to oxidative stress and cell death. Targeting these altered signaling pathways could be considered as the therapeutic approach for neurodegenerative disorders. Ginsenoside Rh1 is known to provide beneficial effects in various diseases such as cancer, diabetes, and inflammation. In this study, human neuroblastoma SH-SY5Y cells were treated with the β-amyloid oligomers alone or in combination with ginsenoside Rh1. We observed that ginsenoside Rh1 was able to attenuate β-amyloid induced oxidative stress and cell death by activating the PI3K/Akt signaling pathway. Based on these findings, we suggest that ginsenoside Rh1 might be an efficacious therapeutic agent for AD.

**Keywords:** Alzheimer's disease; Amyloid-β; Ginsenoside Rh1; SH-SY5Y cells; PI3K/Akt





## 1. Introduction

Dementia can be caused by a wide variety of conditions and a chronic disease with a significant emotional and physical burden [1]. Alzheimer's disease (AD) is a progressive and irreversible neurodegenerative disorder associated with memory deficits and loss of comprehension, language, attention, and judgment [2]. It is the most common cause of dementia affecting people over 65 years of age and is estimated to have up to 75 million cases by 2030 [3]. Despite decades of research and considerable pharmaceutical industry efforts to develop treatments for AD, there are currently no effective treatments for AD [4].

Alzheimer's disease is characterized by the accumulation of insoluble β-amyloid plaques formed due to an improper cleavage of amyloid precursor protein (APP) in addition to the formation of neurofibrillary tangles formed due to the hyper-phosphorylation of micro-tubular protein tau [5,6]. Both of them are involved in neuronal cell death, a prominent feature of AD [7]. Several different signaling pathways were found to be involved in cell survival, and the phosphatidylinositol 3-kinase (PI3K)/protein kinase B (AKT) is one of them. P13K/Akt signaling is generally involved in cell survival, growth, proliferation, and migration. Recent studies demonstrated that inhibition of P13K/Akt by the β-amyloid could cause oxidative stress and neuronal cell death [8,9].

Glycogen synthase kinase-3 (GSK-3) is a protein kinase that phosphorylates various substrates in cellular metabolism [10,11]. It has two isoforms: GSK-3α and GSK-3β [12]. The GSK-3β was found to involved informing the integral component of neurofibrillary tangles (NFT) [13]. The phosphorylation of GSK-3β inhibits the GSK-3β, which is reversibly associated with MAPK signaling [14]. Moreover, p38-MAPK is known to produce neuroinflammatory responses, as well as causing neuronal cell death [15]. Also, recently reported

studies indicated the role of GSK-3 and MAPK in hyperphosphorylation of tau [16,17]. Therefore, targeting these signaling pathways could be serving as a therapeutic agent in AD.

Ginseng (*Panax ginseng*) is a perennial herb belonging to the family Araliaceae that has been used for medicinal purposes since ancient times [18]. Depending on the processing method used, ginseng can be categorized into fresh and white ginseng, red ginseng, and black ginseng [19,20]. It is a natural substance that has been used for more than 2000 years in Korea, East Asia, China, and Japan [21]. Ginseng contains ginsenosides, phenolic compounds, polyacetylenes, alkaloids, polysaccharides, peptides, amino acids, fatty acids, and vitamins [19]. The main ingredient, ginsenoside, is a saponin component that can be active in the cell nucleus, inside the cell, and in the cell membrane [22]. It is also known to improve cognitive ability [23] and memory [24], as well as have anti-aging [25], anti-cancer [26], anti-inflammatory [27], and anti-diabetic [28] effects. It is classified as a protopanaxadiol (PPD) and protopanaxatriol (PPT) according to its structural characteristics [29]. PPD saponins include ginsenoside-Rb1, -Rb2, -Rc, -Rd, and -Rg3. PPT saponins have ginsenoside-Re, -Rg1, -Rg2, and -Rh1. Ginsenoside-Rg2, -Rg3, -Rh1, and -Rh2 are the only ingredients in red ginseng [30]. One of these ginseng saponins, ginsenoside Rh1, is a metabolite of ginsenoside Rg1 caused by bacteria [31,32]. Rg1 has been found to have anti-cancer [33,34], anti-oxidant [35], anti-inflammatory [18], and anti-obesity [36] effects. However, only a few studies have been conducted on the effects of ginsenoside on dementia. The aim of our study was to investigate the neuroprotective effects of ginsenoside Rh1, together with finding out the mechanism through which it could inhibit Aβ-induced oxidative stress and neuronal cell death.

## 2. Materials and Methods

### 2.1. Reagents

Ginsenoside Rh1 ($C_{36}H_{62}O_9$), Aβ$_{1-42}$ peptide, 1,1,1,3,3,3-hexafluoro-2-propanol (HFIP), Fluorescein diacetate (FDA), propidium iodide (PI), and carboxy-H$_2$DCF-DA were obtained from Sigma-Aldrich (St. Louis, MO, USA). Hoechst 33,342 was obtained from Thermo Fisher Scientific (Manassas, VA, USA). Phospho-protein kinase B (Akt; Ser473) antibody, total Akt antibody, phospho-GSK-3β (Ser9) antibody, total GSK-3β antibody, phospho-p38 mitogen-activated protein kinase (MAPK; Thr180/Tyr182) antibody, and total p38 MAPK antibody were purchased from Cell Signaling Technology (Danvers, MA, USA). LY294002 was obtained from Bioscience (Franklin Lakes, NJ, USA), and PD98059 was obtained from Calbiochem (San Diego, CA, USA).

Aβ oligomers were prepared as previously described [37]. The Aβ$_{1-42}$ peptide was dissolved in HFIP to make 1 mM of Aβ monomer. Aliquots (100 μL) of Aβ monomer solution were evaporated completely in a hood for 12 h and stored at −80 °C. They were dissolved in 11 μL of dimethyl sulfoxide (≥99.9 %) and 540 μL of phosphate-buffered saline (PBS) and then stored for 24 h (at 4 °C) to make oligomers. We used 1~4 μM of Aβ oligomers.

### 2.2. Cell Culture and Measurement of Cell Viability

The human neuroblastoma SH-SY5Y cells were obtained from the ATCC (American Type Culture Collection, Manassas, VA, USA). Cells were cultured in Dulbecco's Modified Eagle's medium (10% fetal bovine serum and 1% antibiotic-antimycotic solution) at 37 °C with 5% CO$_2$. To further evaluate the possible mechanisms, phosphoinositide 3-kinase (PI3K; LY294002, 10 μM) inhibitors were treated for 24 h. Each experiment was repeated three times.

Cell Counting Kit-8 assay (CCK-8 assay, Dojindo Molecular Technologies, Rockville, MD, USA) was used to measure cell viability. Briefly, SH-SY5Y cells were seeded into 96-well plates ($1 \times 10^4$ cells/mL). The Aβ oligomers (0.25, 0.5, 1, 1.5, 2, and 4 μM) and different concentrations of Rh1 (5, 10, 20, and 40 μM) with Aβ oligomers (1 μM) were incubated at 37 °C with 5% CO$_2$ for 24 h. After incubation, the CCK-8 assay solution

was added (for 2 h) to each well with protection from light. The absorbance (450 nm) was measured using an Epoch microplate spectrophotometer (BIOTEK Inc., Winooski, VT, USA).

### 2.3. FDA/PI and Hoechst Staining

To measure cell viability, a quantitative evaluation of viable cells using FDA and propidium iodide (PI) double staining was performed. Cells were examined after incubation with FDA (10 μg/mL) and PI (5 μg/mL) for 15 min according to the protocol (ibidi GmbH, Munich, Germany). The staining solution was removed and washed with PBS. The light was blocked and incubated at room temperature for 5 min. Images were obtained using an ultraviolet light microscope and compared under phase-contrast microscopy (Nikon, Tokyo, Japan).

Hoechst staining was used to distinguish the compact chromatin of apoptotic nuclei. SH-SY5Y cells were seeded in 6-well plates ($2 \times 10^5$ cells/mL) and incubated for 24 h. Cells in 6-well plates were exposed to Rh1 (20 and 40 μM) with 1 μM of Aβ oligomers. After 24 h, cells were washed with PBS and fixed once with 10% formaldehyde (Thermo Fisher Scientific, Waltham, MA, USA). The cell membrane was permeabilized with formaldehyde (4 %) and Triton X-100 (0.1%, Sigma-Aldrich, St. Louis, MO, USA) for 15 min. Cells were stained with 5 μg/mL Hoechst 33,342 (Thermo Fisher Scientific, Waltham, MA, USA) for 5 min at 4 °C. The cells were washed once with PBS and analyzed with a fluorescence microscope (Nikon, Tokyo, Japan).

### 2.4. Reactive Oxygen Species (ROS) Assay

SH-SY5Y cells ($2 \times 10^5$ cells/mL) were washed once with PBS and incubated with carboxy-$H_2$DCF-DA (5 μM, Sigma-Aldrich, St. Louis, MO, USA) at 37 °C for 30 min. Then, they were washed 3 times with PBS and measured using a GloMax Discover Microplate Reader (Promega, Madison, WI, USA) at wavelengths of 480 nm excitation and 520 nm emission. Images were obtained using a fluorescence microscope (Nikon Instruments, Tokyo, Japan). The relative fluorescence intensity in SH-SY5Y cells was observed, and cells without treatment were used as the control.

### 2.5. Immunoblot Analysis

The plate was washed once with PBS, scraped with a cell scraper, and centrifuged at 4 °C and 13,000 rpm for 3 min to remove suspended solids. Then, a pro-prep reagent (iNtRON Biotechnology, Gyeonggi-do, Korea) was added. Protein quantification was performed using the PRO-MEASURE kit (iNtRON Biotechnology, Gyeonggi-do, Korea). Equal amounts of protein were separated by 10% SDS-polyacrylamide gel electrophoresis and followed by electrophoretic transfer (Bio-Rad Laboratories, Hercules, CA, USA) onto a polyvinylidene difluoride membrane (Merck Millipore, Bedford, MA, USA). The transferred membrane was blocked in 5% skim milk diluted in Tris-buffered saline solution (TBST). After blocking, the slides were washed with TBST three times for 10 min, and the primary antibody was dissolved in 5% bovine serum albumin overnight at 4 °C. After washing the membrane, a horseradish peroxidase-conjugated secondary antibody (Promega, Madison, WI, USA) was reacted at room temperature for 1 h. The membrane was exposed to an enhanced chemiluminescence solution (iNtRON Biotechnology, Gyeonggi-do, Korea). Expression was confirmed using ImageQuant LAS 500 (GE Healthcare Life Sciences, Uppsala, Sweden).

### 2.6. Statistical Analyses

All data are expressed as the mean and standard error (SE). The statistical analysis was performed using a one-way analysis of variance (ANOVA) test and the Tukey test to verify the significance (*p*-value of <0.05 was considered to be statistically significant).

## 3. Results

### *3.1. Ginsenoside Rh1 Attenuates Aβ Oligomers Neurotoxicity in SH-SY5Y Cells*

SH-SY5Y cells were treated with different concentrations of Rh1 (6.25 μM~100 μM) for 24 h, and cell cytotoxicity was not detected in SH-SY5Y cells (Figure 1a). Next, the concentration of Aβ was screened by treating the cells with different concentrations of Aβ oligomers (0.25~4 μM) for 24 h. It was found that Aβ was able to induce cytotoxicity in a concentration-dependent manner (Figure 1b).

Additionally, to determine the neuroprotective effects of ginsenoside Rh1, SH-SY5Y cells were treated with Aβ (1 μM) and different concentrations of ginsenoside Rh1 (5, 10, 20, and 40 μM) for 24 h. The cell viability analysis showed that ginsenoside Rh1 was able to inhibit Aβ-induced cell death in SH-SY5Y cells (Figure 1c).

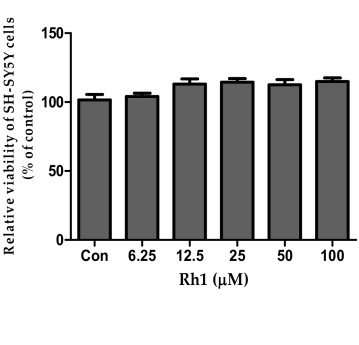
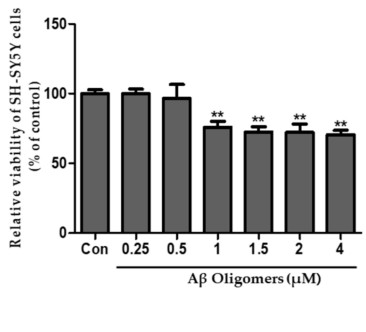
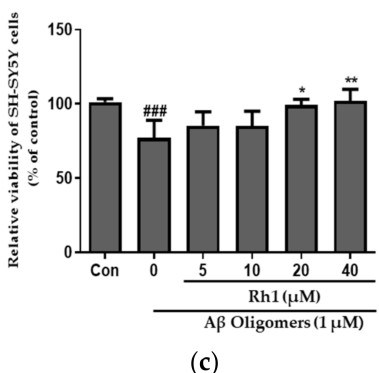

(**a**)  (**b**)  (**c**)

**Figure 1.** Effects of ginsenoside Rh1 on SH-SY5Y cells. (**a**) Ginsenoside Rh1 was used at concentrations of 6.25, 12.5, 25, 50, and 100 μM for 24 h. (**b**) Aβ oligomers-induced neuronal cell death in a dose-dependent manner in SH-SY5Y cells. SH-SY5Y cells were treated with Aβ oligomers at different concentrations for 24 h. ** $p < 0.01$ vs. Controls (Con). (**c**) Ginsenoside Rh1 inhibited Aβ oligomer induced neuronal cell death in SH-SY5Y cells. ### $p < 0.001$ vs. Con, * $p < 0.05$, and ** $p < 0.01$ vs. Aβ oligomers (Aβ). A CCK-8 assay was used to measure cell viability after treatment with ginsenoside Rh1 or/and Aβ oligomers. All data are presented as the mean ± SD, and the experiments were performed at least three times.

### *3.2. Ginsenoside Rh1 Attenuates Intracellular ROS of Aβ Oligomers in SH-SY5Y Cells*

Recent studies revealed that reactive oxygen species are critical factors in the pathogenesis of Alzheimer's disease. After studying the cytoprotective effects of Rh1, we tried to find out the anti-oxidant effects of Rh1 in Aβ-treated SH-SY5Y cells. We found that Aβ oligomers increased the number of PI-positive cells, which is shown as a solid red fluorescence in necrotic cells. It also decreased FDA-positive cells compared to the control cells. On the other hand, Aβ oligomers with ginsenoside Rh1 treatment showed a decrease in the number of PI-positive nonviable cells compared to Aβ oligomers treatment cells (Figure 2).

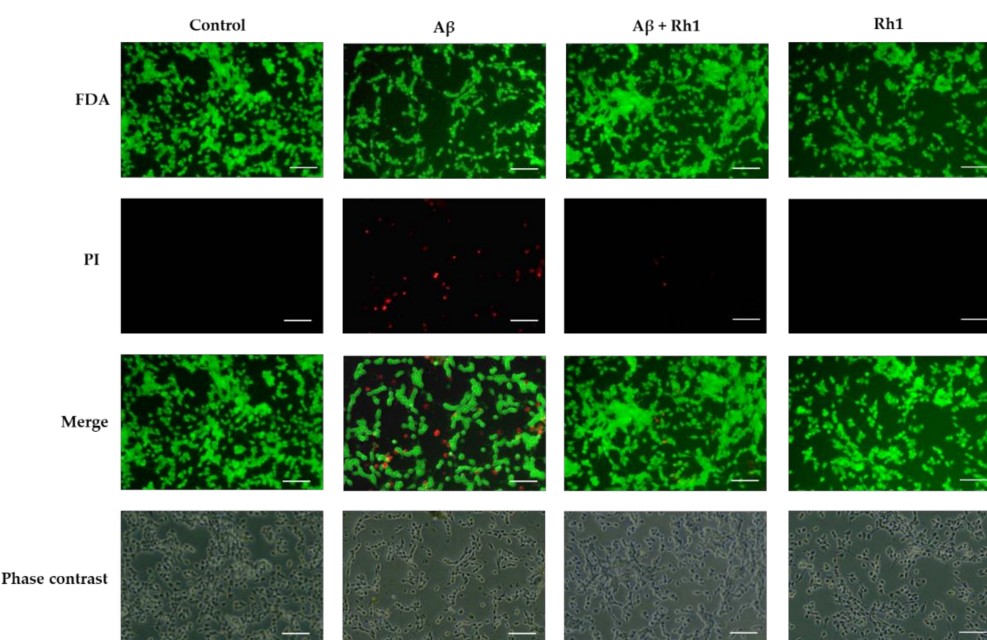

**Figure 2.** Ginsenoside Rh1 attenuated neuronal cell death induced by Aβ oligomers in SH-SY5Y cells. After 24 h of treatment with 1 μM Aβ oligomers or/and 40 μM ginsenoside Rh1, the SH-SY5Y cells were stained with fluorescein diacetate (FDA) and propidium iodide (PI) double staining. PI-positive dead cells (red fluorescence) demonstrated that Aβ oligomers induced neuronal cell death. Scale bar indicates 100 μm.

Furthermore, ginsenoside Rh1 decreased neuronal apoptosis induced by Aβ oligomers, as demonstrated by the decrease in condensed nuclei in the Aβ oligomers with the ginsenoside Rh1 treatment group (Figure 3a). In addition, Aβ oligomers significantly increased the level of intracellular ROS production in SH-SY5Y cells, but Aβ oligomers with the ginsenoside Rh1 treatment group showed decreased ROS levels compared to those of Aβ oligomers (Figure 3b). These results suggested the anti-oxidant properties of ginsenoside Rh1 in Aβ-treated SH-SY5Y cells.

### 3.3. Inhibition of Akt and GSK Pathway in SH-SY5Y Cells

Next, we tried to assess the signaling pathways involved in Aβ-induced SH-SY5Y cells over time. For various durations, 1 μM of Aβ oligomers were exposed to SH-SY5Y cells, and expression levels of phospho-Akt/Akt, phospho-GSK-3β/GSK-3β and phospho-p38 MAPK/p38 MAPK were observed. It was found that 1μM of Aβ was able to down-regulate the phospho-Akt/Akt and phospho-GSK-3β/GSK-3β (Figure 4a,b) and up-regulate the phospho-p38 MAPK/p38 MAPK in a time-dependent manner (Figure 4c). This indicated the role of the PI3K/Akt pathway in Aβ-induced cytotoxicity in SH-SY5Y cells, where Aβ was able to down-regulate phospho-Akt/Akt and phospho-GSK-3β/GSK-3β but up-regulate phospho-p38 MAPK/p38 MAPK.

### 3.4. Cytoprotective Effects of Ginsenoside Rh1 Are Mediated through PI3K/Akt/GSK-3 Pathway

To confirm whether ginsenoside Rh1 attenuated neuronal death via regulation of the Akt and GSK-3 pathways, we used immunoblot analysis. In the co-treated group of ginsenoside Rh1 and Aβ, phospho-Akt/Akt and phospho-GSK-3β/GSK-3β were found to up-regulate compared to the Aβ oligomers only treatment (Figure 5a,b). Also, phospho-p38 MAPK/p38 MAPK was found to down-regulate as compared to the Aβ-alone treated group, indicating that ginsenoside Rh1 might mediate its effects through the activation phospho-Akt/Akt and phospho-GSK-3β/GSK-3β and deactivation of phospho-p38 MAPK/p38 MAPK signaling pathways (Figure 5c).

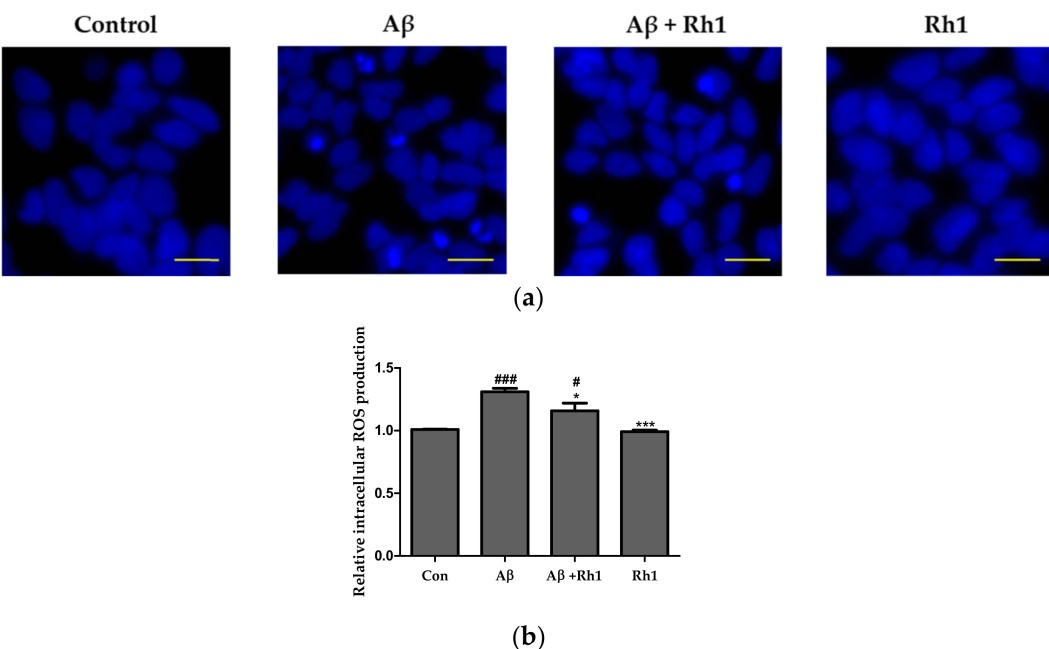

**Figure 3.** Ginsenoside Rh1 attenuated neuronal apoptosis induced by Aβ oligomers in SH-SY5Y cells. (**a**) Hoechst staining was used to measure the number of condensed nuclei (Pyknosis) after treatment with Aβ oligomers (1 μM) and/or ginsenoside Rh1 (40 μM) for 24 h in SH-SY5Y cells. (**b**) Intracellular ROS levels after treatment with 1 μM Aβ oligomers and/or 40 μM ginsenoside Rh1 for 24 h in SH-SY5Y cells. # $p < 0.01$ and ### $p < 0.001$ vs. Control (Con), * $p < 0.05$, and *** $p < 0.001$ vs. Aβ oligomers (Aβ). All data are presented as the mean ± SE, and the experiments were performed at least three times. Scale bar indicates 20 μm.

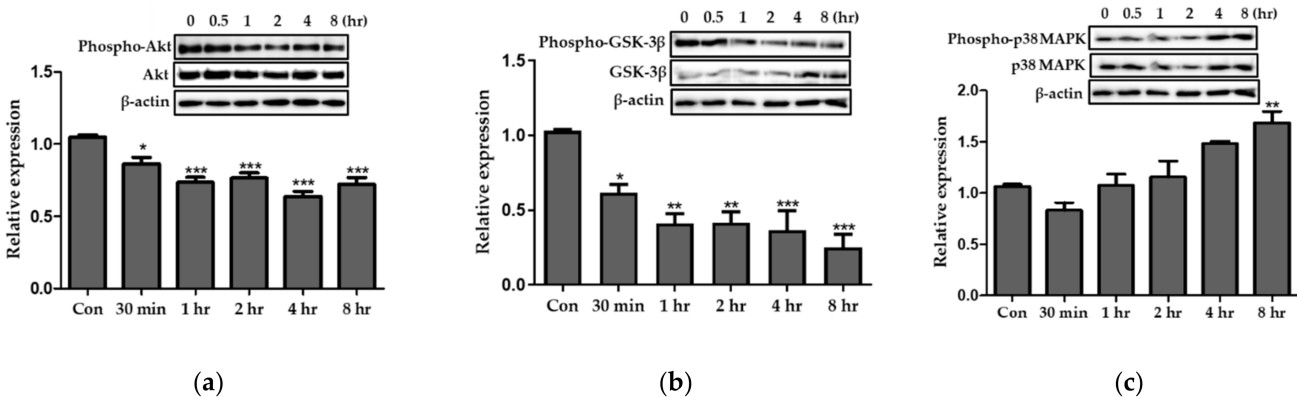

**Figure 4.** Aβ oligomers inhibit the Akt and GSK-3 pathways and activate the p38 MAPK pathway in SH-SY5Y cells. For various time durations, Aβ oligomers (1 μM) were used on SH-SY5Y cells. (**a**) Phospho-Akt (Ser473), (**b**) Phospho-GSK-3β (Ser9), and (**c**) Phospho-p38 MAPK (Thr180/Tyr182). All data are presented as the mean ± SE, and the experiments were performed at least three times. * $p < 0.01$, ** $p < 0.05$, and *** $p < 0.001$ vs. Control (Con).

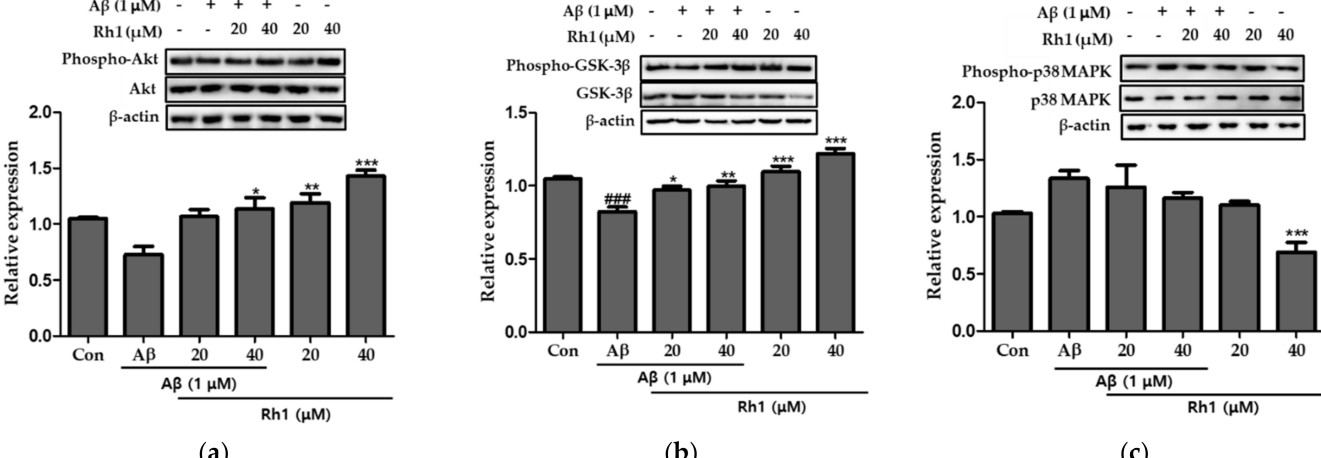

(**a**)             (**b**)             (**c**)

**Figure 5.** Ginsenoside Rh1 attenuated neuronal cell death by Aβ-induced apoptosis in SH-SY5Y cells. (**a**) Phospho-Akt (Ser473), (**b**) Phospho-GSK-3β (Ser9), and (**c**) Phospho-p38 MAPK (Thr180/Tyr182). All data are presented as the mean ± SE, and the experiments were performed at least three times. ### $p < 0.001$ vs. Control (Con), * $p < 0.01$, ** $p < 0.05$, and *** $p < 0.001$ vs. Aβ oligomers (Aβ).

### 3.5. Ginsenoside Rh1 Inhibits Cell Death through the Activation of PI3K/Akt Induced by Aβ Oligomers in SH-SY5Y Cells

To confirm our hypothesis, we further examined the regulation of signaling pathways by ginsenoside Rh1 and Aβ oligomers in the presence of the PI3K/Akt pathway inhibitor (LY294002). It was found that in the presence of the LY294002 (10 μM), Rh1 was unable to show its cytoprotective effects (Figure 6). These results indicated that all the effects of Rh1 observed in Aβ-treated SH-SY5Y cells are mediated through the PI3K/Akt pathway.

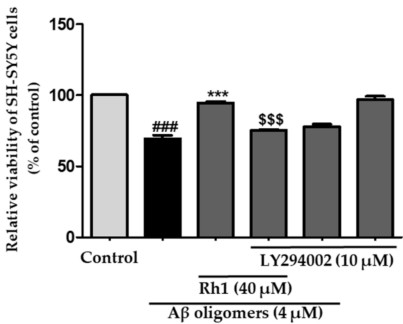

**Figure 6.** Ginsenoside Rh1 attenuated Aβ oligomers-induced neuronal loss, and LY294002 eliminated the protective effects of ginsenoside Rh1 in SH-SY5Y cells. Aβ oligomers (4 μM) were added to SH-SY5Y cells with LY294002 (10 μM) and ginsenoside Rh1 for 24 h. After incubation, cell viability was measured using a CCK-8 assay. All data are presented as the mean ± SE, and the experiments were performed at least three times. ### $p < 0.001$ vs. Control (Con), *** $p < 0.001$ vs. Aβ oligomers, and $$$ $p < 0.001$ vs. Aβ oligomers & ginsenoside Rh1 (Rh1).

## 4. Discussion

The neuroprotective effect of ginsenosides against AD have been reported before [38–42]. Our study reported for the first time that ginsenoside Rh1 significantly attenuates Aβ oligomers-induced oxidative stress and cell death in SH-SY5Y cells. In addition, we further demonstrated that ginsenoside Rh1 protects against Aβ oligomers-induced cell death by activating the PI3K/Akt/GSK-3 pathway.

ROS overproduction is associated with various chronic and degenerative diseases, particularly cancers and neurodegenerative diseases [43]. Various neurodegenerative disorders such as Parkinson's disease, Huntington's disease, and AD may result from biochemical alterations in molecular components due to oxidative stress [44]. The ox-

idative imbalance that leads to neuronal damage may play a central role in AD, and Aβ oligomers and their assemblies could instigate neuronal damage by chemically generating ROS [45,46]. The mitochondrial respiratory chain is a significant site of ROS production in cells, and mitochondrial dysfunction is an essential factor involved in the pathogenesis of AD [47]. Therefore, targeting ROS could be a therapeutic approach in treating various neurodegenerative diseases.

Ginsenoside Rg1, one of the main constituents of *Panax ginseng*, was found to be metabolized to 20 (S)- protopanaxatriol via ginsenosides Rh1 and F1 by gut microbiota [48]. Ginsenoside Rh1 is known to exhibit potent anti-aging, anti-inflammatory, antioxidant, immunomodulatory, and positive effects on the nervous system [49]. Also, Rh1 was found to inhibit hydrogen peroxide-induced ROS generation and subsequent cell death in rat primary astrocytes [50]. Moreover, ginsenoside Rh1 strongly inhibited lipopolysaccharide-stimulated intracellular ROS production in RAW 264.7 cells and suppressed NO and ROS in interferon-gamma-stimulated BV2 microglial cells [51,52]. In addition to that, treatment with ginsenoside Rh1 significantly improved learning and memory ability by promoting cell survival in a mouse model [53]. Additionally, in our study, we found that ginsenoside Rh1 was able to down-regulate the Aβ oligomers-induced oxidative stress.

The PI3K/Akt signaling pathway plays critical regulatory roles in neuroprotection and apoptosis by mediating ROS production [54]. Also, activation of MAPK is known to produce neuroinflammatory responses, as well as cause neuronal cell death [15]. Moreover, recently reported studies indicated the role of GSK-3 and MAPK in hyperphosphorylation of tau, leading to neuronal loss [16,17]. Previous studies have reported that Aβ oligomers can induce apoptosis in various cell types [55,56]. Therefore, we investigated the PI3K/Akt signaling pathways together with studying GSK-3β and MAPK signaling. We found that the PI3K/Akt, GSK-3, and MAPK pathways were altered by Aβ oligomers that could be involved in the cell death, and the ginsenoside Rh1 was found to attenuate Aβ oligomers-induced inhibition of phospho-Akt/Akt and phospho-GSK-3β/GSK-3β together with inhibiting the Aβ oligomers-induced phospho-p38 MAPK/p38-MAPK. Moreover, the PI3K/Akt pathway inhibitor, LY294002, significantly removed the neuroprotective effects of ginsenoside Rh1, suggesting the role of the PI3K/Akt signaling pathway in Rh1 mediated neuroprotective effects.

Ginseng has been widely used in the orient for thousands of years, mainly have prescribed for aging and memory degradation [57]. Because it is mostly converted from Rg1 and Re by intestinal metabolites, the ginsenoside Rh1 in ginseng root is relatively rare [41]. Hou [53] found that mice's learning and memory ability was significantly improved caused by feeding the Rh1 for a long time. Lu [58] reported that ginsenoside Rh1 prevents cognitive impairment of the mouse cortex and hippocampus. This study reported only in vitro neuroprotective effects of ginsenoside Rh1 in human neuroblastoma SH-SY5Y cells. However, to the best of our knowledge, this is the first study describing the protective effect of ginsenoside Rh1 in Aβ oligomers-mediated ROS in AD.

In this study, Aβ oligomers inhibited the phospho-Akt/Akt and phospho-GSK-3β/GSK-3β and activated the phospho-p38 MAPK/p38-MAPK pathway in SH-SY5Y cells. However, ginsenoside Rh1 activated the phospho-Akt/Akt and phospho-GSK-3β/GSK-3β. It also inhibited phospho-p38 MAPK/p38-MAPK with attenuating Aβ oligomers-induced death in SH-SY5Y cells. In conclusion, ginsenoside Rh1 attenuated cell death and oxidative stress via activation of the PI3K/Akt/GSK-3β pathway. Based on these findings, we suggest that ginsenoside Rh1 might be efficacious as a functional compound for alleviating dementia.

**Author Contributions:** Investigation, data arrangement, and writing, M.P.; Original draft preparation, M.P., S.-H.K., and H.-J.L.; Conceptualization and supervision, H.-J.L. All authors have read and agreed to the published version of the manuscript.

**Funding:** The Cooperative Research Program of the Center for Companion Animal Research (Project No. PJ01476703), Rural Development Administration, Republic of Korea, supported this work.

**Data Availability Statement:** Not applicable.

**Acknowledgments:** We would like to thank Yeonjae Lee, an undergraduate at Johns Hopkins University, for correcting English grammar and for proofreading.

**Conflicts of Interest:** The authors declare no conflict of interest.

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
