# Peer review of "Ginsenoside Rh1 Exerts Neuroprotective Effects by Activating the PI3K/Akt Pathway in Amyloid-β Induced SH-SY5Y Cells"

_applsci, doi:10.3390/app11125654_

Round 1
Reviewer 1 Report
This manuscript generates some interest in readers. There are several clarifications that authors have failed to provide. The findings are based on findings from one cell line system. SHSY5Y is a pediatric cancer ( neuroblastoma) cell line. Authors do not explain how this is a relavent and appropriate system to test their hypothesis. Conclusions are not well substantiated by the data presented here.
Author Response
Thank you for allowing us to revise the manuscript. We thank the reviewers for the positive assessment of our study and feel that the comments have significantly improved the manuscript. We have included below point-by-point responses to the critiques and indicated the page numbers where the changes are incorporated. We hope that the revised manuscript will be acceptable for publication in Applied Sciences.
This manuscript generates some interest in readers. There are several clarifications that authors have failed to provide. The findings are based on findings from one cell line system. SHSY5Y is a pediatric cancer (neuroblastoma) cell line. Authors do not explain how this is a relavent and appropriate system to test their hypothesis.
à Thank you for having an interest in our manuscript. The neuroblastoma SH-SY5Y cells have been used in numerous studies in studying Alzheimer’s disease.
Aging Cell. 2020 Jan;19(1): e13054. ‘Activation of PKA/SIRT1 signaling pathway by photobiomodulation therapy reduces Aβ levels in Alzheimer's disease models’
Sci Rep. 2020 Jun 8;10(1):9164. ‘The impact of capsaicinoids on APP processing in Alzheimer’s disease in SH-SY5Y cells’
Mol Neurobiol. 2019 Nov;56(11):7355-7367. ‘Cholinergic Differentiation of Human Neuroblastoma SH-SY5Y Cell Line and Its Potential Use as an In vitro Model for Alzheimer’s Disease Studies’ ‘The human neuroblastoma cell line SH-SY5Y is frequently used as an in vitro model for neurodegenerative disease studies. SH-SY5Y cells are derived from the sympathetic nervous system and considered to be derived from a neuronal lineage in its immature stage. This cell line is characterized by continuously proliferation, expression of immature neuronal proteins, and low abundance of neuronal markers’
J Mol Med (Berl). 2018; 96(10): 1061–1079. ‘STIM1 deficiency is linked to Alzheimer’s disease and triggers cell death in SH-SY5Y cells by upregulation of L-type voltage-operated Ca2+ entry’
Conclusions are not well substantiated by the data presented here.
à Our manuscript showed that Aβ oligomers inhibited the phospho-Akt/Akt and phospho-GSK-3β/GSK-3β pathways, and activated the phospho-p38 MAPK/p38-MAPK pathway in SH-SY5Y cells. And our results confirmed that ginsenoside Rh1 attenuated Aβ oligomers-induced neuronal loss due to oxidative stress with a PI3K/Akt pathway inhibitor (LY294002).

Reviewer 2 Report
- In figure 2 there is no staining of PI in control, Aβ+Rh1, and Rh1 groups. Is it a problem with image quality or staining, or the authors are saying that there is absolutely no cell death in these groups? This is highly unlikely since even the control cells show some cell death during their normal course and even with a protective compound there can never be zero mortality/necrosis in cells.
- Why did the authors choose 1μM dose for Aβ when the remaining doses (1.5, 2 and 4) have almost similar effects on cell viability (fig 1b)?
- Better resolution images are required for fig 3a Hoechst staining since it’s difficult to see the changes otherwise.
- Can the authors provide the loading controls for all their blots?
- Are the changes in phosphorylation of Akt in response to Aβ not time-dependent? There seems to be no consistent pattern between different time points.
- In fig 5 (a,b,c) the stats do not seem to be in line with the pattern of bands. Do the authors have alternate images for the same?
- The authors have used the inhibitor for PI3K (LY294002 is an inhibitor of PI3K and not the complete PI3K/Akt pathway) to emphasize the involvement of PI3K/Akt pathway yet there is no data for changes in PI3K levels. It’s suggested to take a look at PI3K too.
- A grammatical proofread of the entire manuscript is recommended.
Author Response
Thank you for allowing us to revise the manuscript. We thank the reviewers for the positive assessment of our study and feel that the comments have significantly improved the manuscript. We have included below point-by-point responses to the critiques and indicated the page numbers where the changes are incorporated. We hope that the revised manuscript will be acceptable for publication in Applied Sciences.
In figure 2 there is no staining of PI in control, Aβ+Rh1, and Rh1 groups. Is it a problem with image quality or staining, or the authors are saying that there is absolutely no cell death in these groups? This is highly unlikely since even the control cells show some cell death during their normal course and even with a protective compound there can never be zero mortality/necrosis in cells.
à Thank you for your comment. We agreed with the reviewer's point of view that there would never be zero mortality/necrosis in cells, even with a preventive compound. Living cells were incubated with FDA&PI for 15 min at 37℃, CO2 incubator. After washing with PBS, we obtained the image using an ultraviolet light microscope and compared it under phase-contrast microscopy. We took quantify pictures of the PI-stained control, Aβ, Aβ+ Rh1, and Rh1 groups in our sample. We don't want to conclude that Rh1-treated cells had no mortality because we couldn't see red-stained spots on the photos, as predicted in mortal neuroblastoma cells. Instead, a cluster of red-colored cells was discovered in Aβ and Aβ+ Rh1 groups. As a result, we attempted to present the best possible images demonstrating the protective effect of the ginsenoside compound Rh1.
Another manuscript showed the same results as ours.
We would like to present the example figures for FDA and PI double staining in SH-SY5Y cells (Yu et al., FOOD & NUTRITION RESEARCH, 2017 VOL. 61, 1304678) and MDA-MB-231 cells (https://ibidi.com/img/cms/support/AN/AN33_Live_Dead_staining_with_FDA_and_PI.pdf).
FDA and PI double staining in SH-SY5Y cells
FDA and PI double staining in MDA-MB-231 cells
Why did the authors choose 1μM dose for Aβ when the remaining doses (1.5, 2 and 4) have almost similar effects on cell viability (fig 1b)?
à beta-Amyloid Peptide (1-42) (human) is very expensive so after verifying that 1μM was enough to decrease cell viability, we have used 1μM dose for Aβ for the experiments.
Better resolution images are required for fig 3a Hoechst staining since it’s difficult to see the changes otherwise.
à Thank you for your comment. We adjusted the intensity of the image in figure 3a.
Can the authors provide the loading controls for all their blots?
àThank you for your comment. We changed the image to provide the loading control.
Are the changes in phosphorylation of Akt in response to Aβ not time-dependent? There seems to be no consistent pattern between different time points.
à Thank you for your comment. We changed the western blot image in fig 4a. It can be observed that levels of phospho-Akt were down-regulated in a time-dependent manner compared to Akt after treatment with 1 μM of Aβ oligomers. We performed western blot experiments three times.
In fig 5 (a,b,c) the stats do not seem to be in line with the pattern of bands. Do the authors have alternate images for the same?
à Thank you for your comment. We changed the western blot images for the same in fig 5 (a,b,c).
The authors have used the inhibitor for PI3K (LY294002 is an inhibitor of PI3K and not the complete PI3K/Akt pathway) to emphasize the involvement of PI3K/Akt pathway yet there is no data for changes in PI3K levels. It’s suggested to take a look at PI3K too.
à Thank you for your valuable comment. We’re sorry that we didn’t get a western blot analysis with PI3K. We tried to show the SH-SY5Y cell viability as protective effects of ginsenoside Rh1 or ginsenoside Rh1+Aβ. The LY294002 treatment eliminated the protective effects of ginsenoside Rh1 in SH-SY5Y cells.
A grammatical proofread of the entire manuscript is recommended.
à Thank you for your comment. The manuscript has been revised by a professional, and all language-related errors have been corrected. Please see the attached certificate of English editing.

Round 2
Reviewer 2 Report
The reviewer is still not convinced with the answer to the queries. The authors have not exactly addressed the following points.
Loading control is still missing from the blots
Beta-actin is the same for Figures 4c and 4d
Figure 3 is still not clearly visible
Author Response
We appreciate you for taking the time to review our study and feel that the comments have significantly improved the manuscript. We have responded point-by-point to the reviewer’s questions and comments. We hope that the revised manuscript will be acceptable for publication in Applied Sciences.
Loading control is still missing from the blots
à Thank you for your comment. We added the western blot images of loading control in Figure 5.
Beta-actin is the same for Figures 4c and 4d
à Thank you for your comment. Figures 4B and 4C are the same PVDF membrane. After stripping the membrane, we’ve done the western blot once for the β-actin antibody.
We found the other studies that used the same loading control.
Shlomai et al., Endocrine-Related Cancer (2017) 24, 519–529
Omar et al., Molecular Pharmacology (2009) 76 (5), 957-968
Wang et al., PLoS One (2013) 11;8 (11):e79117.
Wu et al., Stem Cell Research & Therapy (2019) 10, Article number: 311.
Figure 3 is still not clearly visible
à As follow your comment, we have changed the images in figure 3a.
